

# Topology-aware pathway analysis of spatial transcriptomics

Siras Hakobyan[1], Maria Schmidt[2], Hans Binder[2,3] and Arsen Arakelyan[1,4]

[1] Bioinformatics Group, Institute of Molecular Biology NAS RA, Yerevan, Armenia
[2] Interdisciplinary Centre for Bioinformatics, Universität Leipzig, Leipzig, Saxony, Germany
[3] Armenian Bioinformatics Institute, Yerevan, Armenia
[4] Bioengineering, Bioinformatics and Molecular Biology, Russian-Armenian University, Institute of Biomedicine and Pharmacy, Yerevan, Armenia

Corresponding author
Siras Hakobyan,
sirashakobyan@gmail.com

## ABSTRACT

Spatial transcriptomics (ST) has transformed genomics by mapping gene expression onto intact tissue architecture, uncovering intricate cellular interactions that bulk and single-cell RNA sequencing often overlook. Traditional ST workflows typically involve clustering spots, performing differential expression analyses, and annotating results *via* gene-set methods such as overrepresentation analysis (ORA) or gene set enrichment analysis (GSEA). More recent spatially-aware techniques extend these approaches by incorporating tissue organization into gene-set scoring. However, because they operate primarily at the level of individual genes, they may overlook the connectivity and topology of biological pathways, limiting their capacity to trace the propagation of signaling events within tissue regions. In this study, we address that gap by translating gene expression into pathway-level activity using the Pathway Signal Flow (PSF) algorithm. PSF integrates expression data with curated interaction networks to compute numeric activity scores for each branch of a biological pathway, producing a functionally annotated feature space that captures downstream signaling effects as branch-specific activity values. We applied PSF to two public 10x Genomics Visium datasets (human melanoma and mouse brain) and compared clustering based on PSF-derived pathway activities from 40 curated Kyoto Encyclopedia of Genes and Genomes (KEGG) signaling pathways and gene expression with standard Seurat Louvain clustering and spatially aware methods (Vesalius, spatialGE). We observed good correspondence between PSF-based and expression-based clustering when spatially aware clustering methods were used. This suggests that branch-level pathway activities can themselves drive clustering and pinpoint spatially deregulated processes. To assess cluster-specific functional annotation, we compared PSF results to conventional ORA (based on marker genes) and GSDensity (based on cluster-specific gene sets). PSF identified a broader set of significant pathways with substantial overlap with both ORA and GSDensity, providing increased sensitivity due to its branch-level resolution. We further demonstrated that PSF-derived activity values can be used to detect spatially deregulated pathway branches, yielding results comparable to those obtained with spatially aware gene set analysis approaches such as GSDensity and spatialGE. The availability of pathway topology and branch-specific information also enabled the identification of potential intercellular communication *via* ligand-receptor interactions between deregulated pathways in adjacent tumor regions. To support interactive exploration of results, we developed the PSF Spatial Browser, an R Shiny

application for visualizing pathway activities, gene expression patterns, and deregulated pathway networks.

# INTRODUCTION

Spatial transcriptomics (ST) has revolutionized genomics by enabling the study of gene expression within the context of tissue architecture. This innovative approach captures the spatial organization of gene activity, providing insights into cellular microenvironments typically lost in bulk and single-cell RNA sequencing methods (*Marx, 2021*). Understanding the spatial heterogeneity of gene expression is especially critical in diseases such as cancer, where the tumor microenvironment significantly impacts disease progression and therapeutic responses (*Eng et al., 2019*).

ST methods broadly fall into two categories: imaging-based and sequencing-based. Imaging-based methods capture spatially resolved expression data through fluorescence *in situ* hybridization (FISH) or *in situ* sequencing techniques, providing single-cell or subcellular resolution but limited to a select set of targeted genes (*Williams et al., 2022*). In contrast, sequencing-based approaches measure the whole transcriptome from tissue slices, spatially partitioned into discrete cell spots (Visium by 10X Genomics and GeoMx by Nanostring) or single cells (Visium HD by 10X Genomics and Stereo-seq by STOmics) (*Lim et al., 2025*). Currently, 10x Genomics platforms, including the higher-resolution Visium HD, are among the most widely adopted technologies. The standard Visium captures approximately ten cells per spot, whereas Visium HD typically achieves single-cell resolution.

ST data analysis pipelines largely overlap with single-cell RNA sequencing approaches, reflecting similar data structures characterized by high dimensionality and sparsity, especially at single-cell resolution. However, ST data uniquely incorporates spatial positional information for tissue spots. Standard methods include data preprocessing, clustering, visualization tools, and functional annotation approaches such as gene set-based overrepresentation analysis (ORA) and Gene Set Enrichment Analysis (GSEA) (*Draghici et al., 2003*; *Subramanian et al., 2005*).

Recently, methods incorporating spatial information into analysis have been developed to enhance clustering accuracy and interpretability. For instance, the SpatialPCA method conducts spatially aware principal component analysis by embedding gene expression data into a low-dimensional space, preserving spatial correlations among tissue locations (*Shang & Zhou, 2022*). Another innovative method, Vesalius, transforms gene expression data into RGB color codes, employing image-based segmentation to detect spatially relevant clusters (*Martin et al., 2022*). Additionally, GraphST leverages graph neural networks to effectively model spatial relationships between tissue spots (*Long et al., 2023*).

Spatial data is also increasingly utilized to improve functional analyses. Specifically, two approaches have recently been introduced that perform spatially-aware gene set enrichment analyses. SpatialGE detects spatial aggregation of spots expressing particular gene sets (*Ospina et al., 2022*). The GSDensity utilizes multiple correspondence analysis (MCA) to co-embed cells and genes into a shared latent space, estimating pathway activity based on the spatial density of genes within pathways (*Liang et al., 2023*).

Reviewing existing single-cell and ST analysis frameworks revealed a gap in methods specifically tailored for topology-based pathway analyses. Current approaches predominantly rely on gene sets, thus overlooking detailed information about protein interactions, their types, and interaction directions. Our group previously developed the Pathway Signal Flow (PSF) algorithm, which integrates gene expression data with pathway topology information to calculate the activity of individual pathway branches for single samples (*Hakobyan et al., 2023*). We demonstrated its effectiveness in analyzing bulk and single-cell (sc) transcriptomic data, deriving activity values for pathway branches (effector protein activities) linked to distinct biological processes. In scRNA analysis, these activity values can subsequently be utilized for cell clustering and identification of cluster-specific pathway activities. Conducting pathway activity analysis prior to clustering incorporates pathway information into the analytical pipeline, providing functional context that simplifies the interpretation of cellular and cluster-level functional states.

In the present study, we demonstrate the utility of topology-based pathway analysis using the PSF toolkit R package on publicly available 10x Visium spatial transcriptomic datasets. Specifically, we analyze human melanoma and mouse brain datasets and compare our findings to traditional and spatially-aware gene set analysis methods. We demonstrate that pathway activity values effectively integrate with both standard and spatially-aware clustering algorithms, yielding results comparable to established methods. Additionally, we have developed an interactive browser for visualizing pathway activities within spatial transcriptomic data, facilitating exploratory analysis of spatially regulated biological processes.

## MATERIALS AND METHODS

### Data acquisition and preprocessing

The publicly available spatial transcriptomic data of formalin-fixed paraffin-embedded (FFPE) melanoma and fresh frozen mouse brain tissue were obtained from the 10X Genomics open datasets (FFPE Human Melanoma: https://www.10xgenomics.com/datasets/human-melanoma-if-stained-ffpe-2-standard; FF Mouse Brain Serial Section 1 (Sagittal-Anterior): https://www.10xgenomics.com/datasets/mouse-brain-serial-section-1-sagittal-anterior-1-standard-1-1-0). The datasets comprised 3,458 and 2,695 spots, with a median of 7,598 and 6,015 expressed genes per spot.

Gene expression raw counts, spot coordinate information, and tissue scans were imported with the Seurat R package (*Satija et al., 2015*). Gene Expression data was filtered and normalized using the SCT transform method (*Hafemeister & Satija, 2019*).

## Pathway activity analysis

The pathway activity analysis was performed with the SCT-transformed gene expression matrix from the previous step as input data after centralization against the global mean expression. The PSF toolkit package was used to calculate the branch-level activity of signaling pathways (*Hakobyan et al., 2023*). Briefly, gene expression fold-change values were mapped to the corresponding nodes of the pathway. Then, pathway signal propagation along the pathway branches was calculated by either multiplying or dividing node values, depending on the interaction type (multiplication for activation and division for inhibition), starting from the input nodes and progressing through to the terminal (sink) nodes. A detailed description of the PSF algorithm and signal propagation formulas is available in our previous publication (*Nersisyan et al., 2017*).

For the pathway activity analysis, a curated set of 40 Kyoto Encyclopedia of Genes and Genomes (KEGG) signaling pathways containing 709 sink or terminal nodes was used. These pathways were curated using the PSF toolkit's curation module to recover missing information and update the pathways for improved topological analysis (*Kanehisa & Goto, 2000*; *Hakobyan et al., 2023*).

## Overall strategy

To evaluate the performance of PSF-based analysis on spatial transcriptomics datasets compared to other spatially aware and conventional functional analysis approaches, we designed a multi-step strategy applied to two publicly available spatial datasets.

- First, we processed both gene expression and pathway activity data and applied three different clustering methods, one standard (Seurat) and two spatially-aware methods (Vesalius and spatialGE), to identify spatially distinct regions of spots. We then compared the clustering outcomes across different methods and data types (gene expression *vs.* pathway activity) to assess consistency and spatial resolution.
- Cluster assignments from both expression- and pathway-based clustering were used to identify enriched or altered gene sets using two approaches: (1) ORA based on marker genes of each cluster, and (2) cluster-specific pathway analysis using GSDensity, which computes the specificity of each pathway to a cluster based on Jensen–Shannon divergence. Finally, we compared the cluster-specific pathways identified by ORA and GSDensity with the PSF-derived significantly deregulated pathway branches to evaluate overlap, complementarity, and biological relevance.
- We also performed clustering-independent gene set analysis on the gene expression data to identify spatially relevant pathways using two spatially-aware approaches: SpatialGE and GSDensity. In addition, the kernel density estimation algorithm implemented in GSDensity was applied to PSF-derived pathway activity values to detect spatially relevant pathways.

To illustrate the utility of pathway branch-level activity data we investigated potential crosstalk between spatially adjacent tumor regions by analyzing receptor–ligand connections between differentially activated pathways across cluster borders for the human melanoma dataset.

To facilitate exploratory analysis, we developed the PSF Spatial Browser, an interactive R Shiny application for visualizing spatially resolved pathway activities and pathway-level interactions on tissue slices (available at https://github.com/hakobyansiras/psf).

The code to reproduce the results and figures is available in the GitHub repository at https://github.com/hakobyansiras/psf/blob/main/inst/extdata/psf_spatial/psf_spatial_study_reproduction.R.

### Clustering of gene expression and pathway activity data

Gene expression and pathway activity values of both datasets were clustered with three different methods, which include standard Louvain clustering implemented in the Seurat R package and two spatially-aware clustering methods, Vesalius and spatialGE (*Stuart et al., 2019*; *Ospina et al., 2022*; *Martin et al., 2022*). For spatially aware clustering methods, raw expression counts and corresponding spatial coordinates of spots were used to construct Vesalius and STlist objects for downstream analyses. Spatially-aware clustering of gene expression and pathway activity data was then performed using default parameters for each method. Finally, the resulting cluster identities were integrated into the Seurat object for subsequent cluster-specific functional analysis. Clustering results between the data types and clustering algorithms were compared with the *adjustedRandIndex* function from mclust R package.

### Spatially aware gene set and PSF analysis

To compare spatially coordinated pathway activity of PSF and enrichment patterns across the tissue, we applied two spatially-aware gene set analysis methods, spatialGE and GSDensity, to melanoma and mouse brain datasets using 40 KEGG signaling pathways.

The STenrich function from the spatialGE R package was used to perform spatially aware gene set enrichment analysis on SCT-transformed gene expression data with default parameters. All the pathways with FDR-adjusted $p$-value < 0.05 were considered significant.

The GSDensity R package was used to identify spatially relevant gene sets by co-embedding genes and spatial spots into a two-dimensional space using multiple correspondence analysis (MCA) *via* the *compute.mca* function. Gene set density levels were then estimated using kernel density estimation in the MCA space with the *computekldFix* function. To assess significance, $p$-values were calculated by comparing the density of the target gene set with that of randomly constructed gene sets of the same size. Gene sets with an FDR-adjusted $p$-value < 0.05 were selected for downstream analysis.

Activity scores for the selected gene sets were calculated using the *run.rwr.list* function, which applies a random walk with a restart algorithm. These scores were subsequently used to estimate spatial specificity using the *compute.spatial.kld.df* function, which applies kernel density divergence to compare spatial distributions.

Additionally, we used the *compute.spatial.kld.df* function to assess spatial specificity of PSF-derived pathway activity scores by providing pathway activity values directly as input instead of gene expression data.

## Cluster-specific gene set analysis and comparison with the PSF results

Group assignments obtained from three clustering methods, applied separately to gene expression and pathway activity data, were used to assess the performance of functional analysis approaches in detecting cluster-specific deregulated pathways. We compared pathway analysis tools under two threshold settings, representing low and high stringency.

Cluster-specific pathway branches were identified by performing one-versus-all comparisons of sink node log2-transformed pathway activity values. This was done using the Wilcoxon rank-sum test implemented in the *FindMarkers* function of the Seurat R package (*Stuart et al., 2019*). Pathway branches (sink nodes) were considered significantly deregulated if they had a false discovery rate (FDR)-adjusted $p$-value < 0.05 and an absolute log2 fold change > 0.25 (low stringency) or 0.5 (high stringency).

Similarly, for ORA, differentially expressed genes were identified using the same statistical test. Genes with an absolute log2 fold change > 0.5 or 1 and FDR-adjusted $p$-value < 0.05 were selected for enrichment analysis. ORA was performed using the *fgsea* R package (*Sergushichev, 2016*), with 40 curated KEGG signaling pathways used across all functional analysis methods.

Cluster assignments were also used to assess the specificity of gene sets identified by the GSDensity method. This was accomplished using the *compute.spec* function, which calculates cluster-wise specificity (CWS) scores based on Jensen–Shannon divergence. For each cluster and gene set pair, the method compares a binary cluster membership vector with the corresponding gene set activity scores across all spots. Higher CWS values indicate stronger specificity of the gene set to the given cluster. For each cluster, gene sets with specificity scores exceeding the 60th and 80th percentiles of the overall distribution were selected, representing gene sets most relevant to that cluster under low and high stringency conditions, respectively.

### Pathway activity profiles between cluster borderlines

The LinkedDimPlot function from the Seurat package was customized to manually select groups of spots from different clusters directly on the spatial plot. This was used to investigate differences between core and border spots within the same clusters and to examine the interactions of adjacent spots in the border regions of different clusters at the pathway level using linear regression analysis. The FDR-adjusted $p$-value < 0.05 of regression coefficients was considered statistically significant. For adjacent border spot groups, ligand–receptor interactions were identified by matching the sink nodes of significant pathways in one group with the input nodes of significant pathways in the adjacent group.

## RESULTS

### Clustering of expression and PSF-based spatial spots

To explore spatial patterns in gene expression and pathway activity, we performed clustering of both gene expression data and PSF-derived pathway activity values in human melanoma and mouse brain spatial transcriptomics datasets. For this analysis, we applied three

different clustering approaches: the non-spatial Louvain algorithm implemented in the Seurat R package, and two spatially aware methods, Vesalius and spatialGE (*Stuart et al., 2019*; *Ospina et al., 2022*; *Martin et al., 2022*). Clustering results were compared both visually and quantitatively using the adjusted Rand index (ARI) to assess similarity between cluster assignments.

First, we compared clustering results obtained from gene expression and pathway activity data within each method for melanoma. The highest similarity between expression-based and pathway activity-based clustering was found for Vesalius (ARI of 57%), followed by spatialGE (ARI = 41%) and Seurat (ARI = 27%). Visual comparison of clustering results on the tissue spatial maps also revealed an overall correspondence between expression- and pathway activity-based clustering in terms of cluster locations and shapes, although this was more consistent for certain clusters than others (Fig. 1). In the mouse brain dataset, clustering results showed a generally lower similarity between expression- and PSF-based clustering across all methods compared to the melanoma dataset (Vesalius ARI: 34%, SpatialGE: ARI 18%, Seurat: ARI 20%). Again, visual comparison between gene expression- and pathway activity-based clusters on the spatial maps of the mouse brain tissue revealed that Vesalius produced more coherent and spatially consistent clusters, with clearer boundaries and reduced noise while maintaining similar spatial locations across both data types (Fig. 2).

We further compared the clustering consistency between the three clustering approaches with gene expression data for both datasets. In the human melanoma dataset, clustering based on gene expression data showed moderate similarity between Seurat and the two spatially aware methods (left column of the spatial plots in Fig. 1), with an ARI of around 54%. In the mouse dataset, the ARI index was 43% between Seurat and Vesalius and 52% between Seurat and SpatialGE (left column of the spatial plots in Fig. 2).

These results indicate that the clustering of spatial transcriptomic data can vary depending on the algorithms used.

## Comparison of clustering-based functional annotation tools

Next, we used the clustering results to perform functional annotation of identified spot groups using three different functional analysis approaches. First, ORA was applied to marker genes of each cluster, identified through differential gene expression analysis. Second, the GSDensity method was used to detect spatially enriched gene sets based on spatially coordinated gene set activity patterns (see 'Methods'). Third, to identify deregulated pathway branches from PSF-derived pathway activity data, we applied the Wilcoxon rank-sum test to detect significantly activated or inhibited sink nodes (terminal pathway nodes) for each cluster compared to all others (see 'Methods'). For both ORA and GSDensity, we used the same set of 40 curated KEGG signaling pathways as input gene sets, consistent with the pathways analyzed using the PSF toolkit.

The results showed that PSF analysis identified a substantially higher number of deregulated pathways in all clusters compared to ORA and GSDensity (Fig. 3). This can be attributed to the advantage of PSF in evaluating activity changes at the level of individual pathway branches, rather than assessing the pathway as a whole. In PSF-based

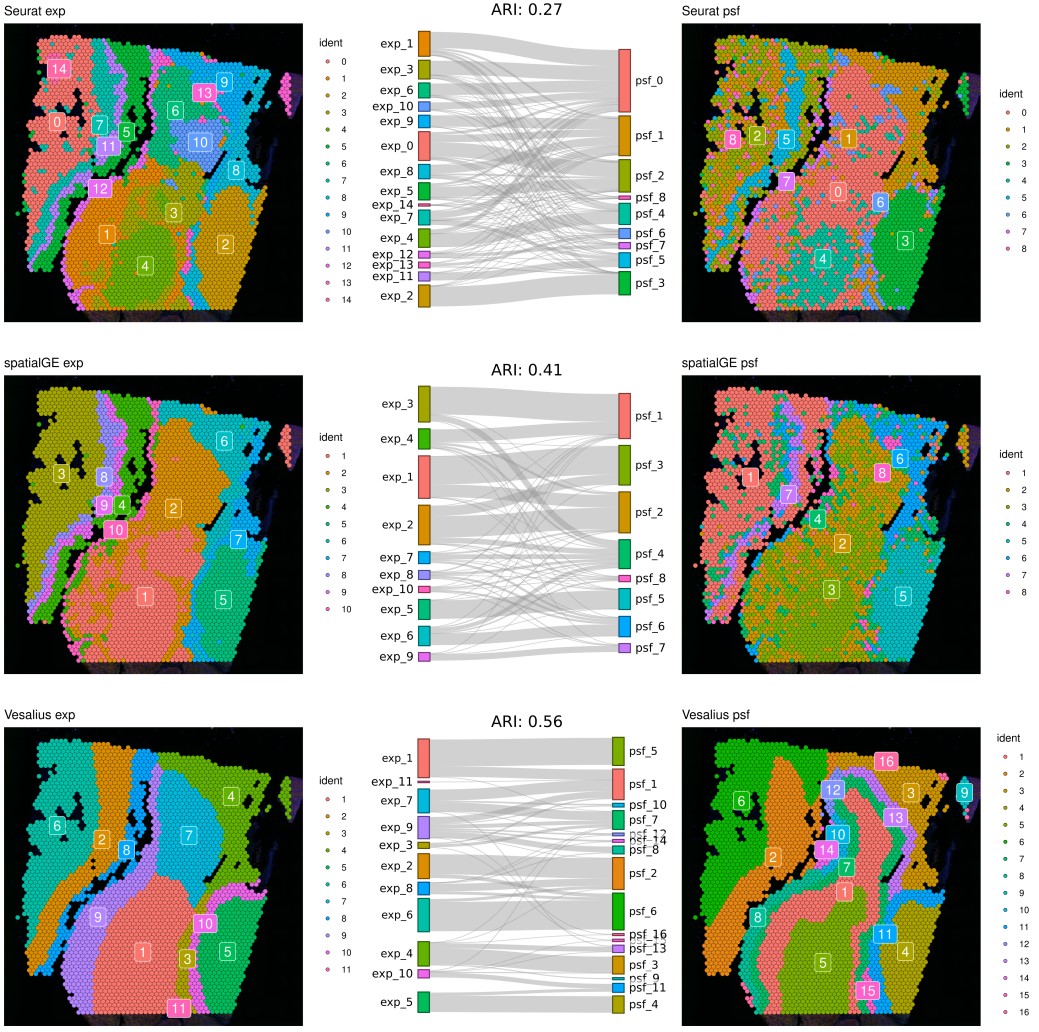

**Figure 1 Comparative clustering of the human melanoma tissue dataset.** Clustering results were obtained using three different algorithms, visualized on the human melanoma tissue slice. The left column shows clusters based on gene expression data, while the right column displays clusters derived from PSF pathway activity values. Sankey diagrams between each pair of plots illustrate the overlap between expression-based and pathway activity-based clustering results. Adjusted Rand index (ARI) values indicating the similarity between cluster assignments are shown above each Sankey diagram. Among the methods tested, the Vesalius clustering approach demonstrated the highest similarity between gene expression-based and pathway activity-based clustering.

analysis, a pathway was considered significantly altered if at least one of its branches showed a significant activity change within a cluster compared to the rest of the tissue.

To further support this observation, we calculated the total activity change per pathway (defined as the sum of absolute log2 pathway activity values) and compared these values between pathways identified by PSF alone and those overlapping with ORA and/or GSDensity. Consistently, pathways that overlapped between PSF and other methods demonstrated higher total activity changes across many clusters (Fig. S1). Furthermore,

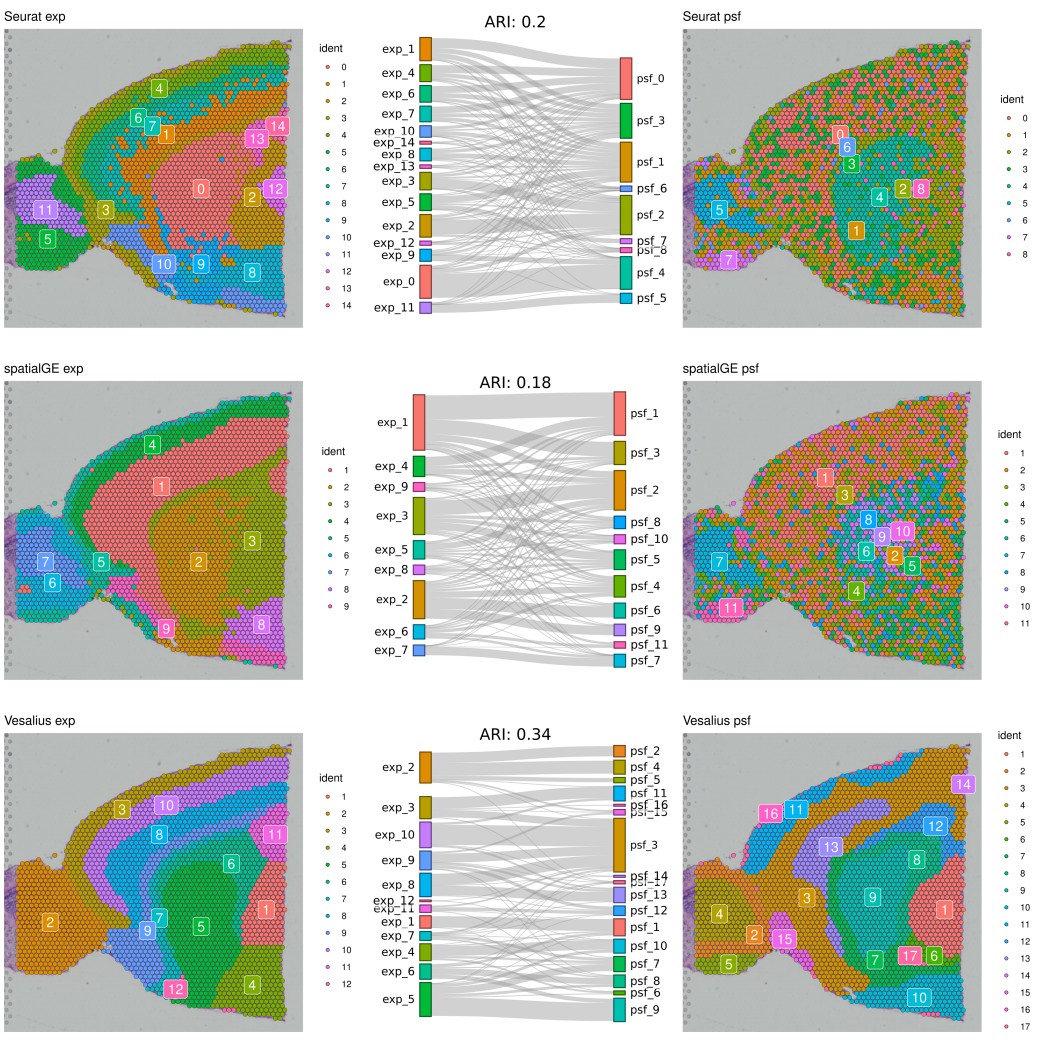

**Figure 2** **Comparative clustering of the mouse brain tissue dataset.** Clustering results were obtained using three different algorithms, visualized on the mouse brain tissue slice (Sagittal-Anterior). The left column shows clusters based on gene expression data, while the right column displays clusters derived from PSF pathway activity values. Sankey diagrams between each pair of plots illustrate the overlap between expression-based and pathway activity-based clustering results. Adjusted Rand index (ARI) values indicating the similarity between cluster assignments are shown above each Sankey diagram. Among the methods tested, the Vesalius approach showed the highest correspondence between gene expression and pathway activity-based clustering.

that PSF captures functionally relevant but potentially subtle changes at branches that might be missed when performed total pathway level analysis.

In most cases, the majority of pathways detected by ORA or GSDensity overlapped with those identified by PSF analysis (Fig. 3). The overlap between PSF and GSDensity was generally higher than between PSF and ORA. The lowest overlap was observed between GSDensity and ORA, likely reflecting the differences in their methodological approaches, with GSDensity focusing on the spatial coordination of gene sets, and ORA relying on enrichment of pre-defined gene lists.

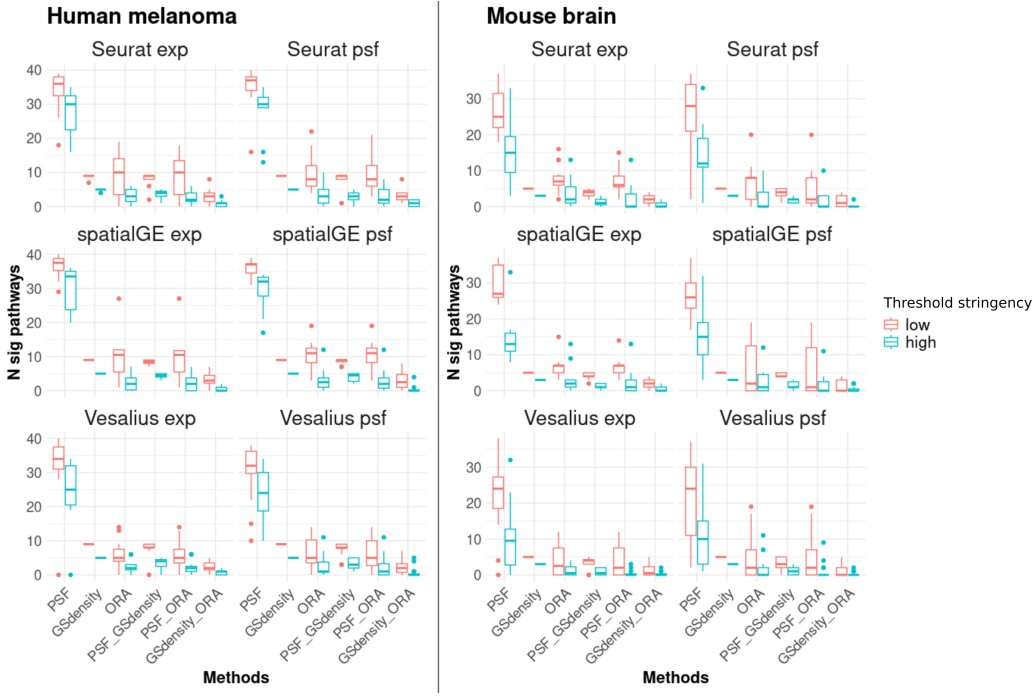

**Figure 3  Number of significantly deregulated pathways detected by each of the three functional annotation methods and their pairwise overlaps under two threshold settings.** Low stringency criteria are defined as follows: for ORA, absolute $\log_2$ fold change greater than 0.5 for marker genes; for PSF, absolute $\log_2$ activity fold change greater than 0.25 for pathway sink nodes; and for GSDensity, cluster specificity scores above the 60th percentile. High stringency criteria are: absolute $\log_2$ fold change greater than 1, absolute $\log_2$ activity fold change greater than 0.5, and cluster specificity scores above the 80th percentile. The left two columns of panels show results for the human melanoma dataset, and the right two columns show results for the mouse brain dataset. Each row corresponds to one clustering algorithm, with gene expression–based clustering on the left and pathway activity–based clustering on the right. Across both datasets and threshold settings, PSF consistently detects a higher number of significant pathways compared to the other methods. Furthermore, there is a greater overlap between pathways detected by PSF and GSDensity than those shared with ORA.

To assess the robustness of these findings, we repeated the comparisons using both low- and high-stringency thresholds (see 'Methods'). The overall trends of pathway overlaps remained consistent regardless of the threshold applied (Fig. 3), indicating that the observed differences between the methods were not strongly influenced by threshold settings. Detailed cluster-specific pathway overlaps for high stringency thresholds visualized by Venn diagrams are provided in Document S2.

As a representative example, we examined pathway alterations and their overlaps within one of the most conserved clusters identified by all clustering methods and data types —a region located in the bottom right corner of the human melanoma tissue slice (Fig. 1). This region corresponds to a highly proliferative tumor area classified as proliferative melanoma by *Schmidt et al. (2024)*. In this region, five pathways, cAMP, p53, MAPK, VEGF, and Insulin signaling were consistently identified as significantly altered by both PSF and GSDensity. Additionally, ORA detected the IL-17 signaling pathway as commonly

deregulated together with PSF. All of these pathways are well-known to be involved in melanoma development, progression, or therapeutic resistance, as supported by multiple studies (*Sullivan & Flaherty, 2013*; *Bang & Zippin, 2021*; *Loureiro et al., 2021*; *Váraljai et al., 2023*; *Malekan et al., 2024*).

Interestingly, the first most significant pathway identified by PSF, but not detected by ORA or GSDensity, was the Hippo signaling pathway a key regulator of organ growth which is deregulated in many cancers (*Lv & Zhou, 2023*). This pathway is frequently deregulated in melanoma, with YAP1, a hub node in the Hippo pathway, playing a crucial role in promoting cell proliferation and metastasis in melanoma (*Zhang et al., 2020*). The deregulation of this pathway was primarily driven by changes in YAP1 expression and inhibition of upstream YAP1 inhibitors (Fig. S2). This example highlights the advantage of topology-aware analysis provided by PSF in detecting biologically important pathways overlooked by gene set-based approaches.

As a second example, we examined a specific region of spots in the mouse brain dataset consistently identified across clustering methods, located in the olfactory bulb region of the mouse brain located in the left side of the tissue slice (Fig. 2). In this region, the Wnt signaling pathway was identified by all three methods (PSF, ORA, and GSDensity). Wnt signaling is well-known to regulate neurogenesis in the olfactory system, particularly in olfactory stem cells (*Wang et al., 2011*). Other pathways commonly detected by both PSF and ORA included calcium signaling, chemokine signaling, and cAMP signaling, all of which are implicated in regulating olfactory neuron function and development (*Menini, 1999*; *Boccaccio, Menini & Pifferi, 2021*; *Senf et al., 2021*). Interestingly, in this region, GSDensity detected fewer pathways compared to ORA, which contrasts with the general trend observed in other clusters.

A list of cluster-specific deregulated pathways identified using high stringency thresholds for all six clustering approaches is provided in Datasets 1 and 2, corresponding to the human melanoma and mouse brain datasets, respectively.

## Evaluation of spatially aware pathway analysis methods with psf-based approach

As observed across different clustering algorithms, including both spatially aware and conventional methods, there are considerable differences in the number of identified clusters, their sizes, and spatial distributions. These discrepancies emphasize that clustering-based functional or pathway analysis can produce inconsistent results, heavily influenced by the choice of the clustering method and its parameters.

In parallel to traditional clustering or experimental group-based annotation approaches, the availability of spatial coordinates in spatial transcriptomics datasets has enabled the development of spatially aware gene set analysis methods. To the best of our knolwdge currently available spatially aware functional analysis methods are spatialGE and GSDensity that utilize spatial information to detect gene sets exhibiting spatial aggregation or coordination (*Ospina et al., 2022*; *Liang et al., 2023*).

In this study, we applied both GSDensity and spatialGE to identify spatially relevant gene sets and compared their results to spatially deregulated pathways inferred using pathway

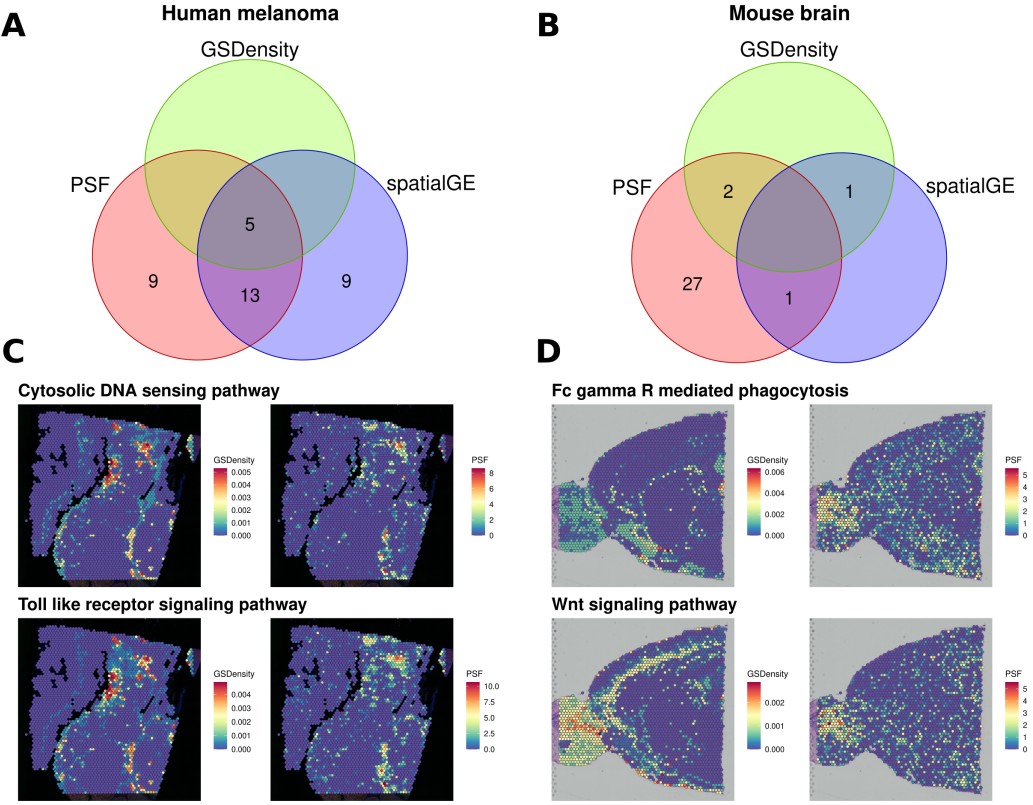

**Figure 4** **Comparison of spatially aware pathway analysis methods.** (A, B) Venn diagrams showing the overlap among pathways identified as spatially coordinated by two gene set–based methods (spatialGE and GSDensity) and by the PSF toolkit's spatial pathway activity for (A) the human melanoma dataset and (B) the mouse brain dataset. (C, D) Spatial visualization of pathway relevance scores and PSF-derived activity values for the set of pathways detected by both GSDensity and PSF. (C) shows results for the melanoma tissue slice, and (D) for the mouse brain slice. PSF-derived activity values below zero were set to zero to match the nonnegative scale of GSDensity scores and to highlight regions of pathway upregulation on the tissue.

activity values from the PSF toolkit. For this purpose, we used gene expression and spatial coordinate data from human melanoma and mouse brain datasets. After identifying spatially enriched gene sets with GSDensity and spatialGE, we applied the kernel density estimation approach from the GSDensity package to the PSF-derived pathway activity values to detect spatially deregulated signaling pathways.

We then compared the results of all three approaches (Fig. 4). In the melanoma dataset, five pathways were consistently identified as spatially deregulated by all three methods: Cytosolic DNA sensing, NF-kappa B signaling, PPAR signaling, TNF signaling, and Toll-like receptor signaling (Fig. 4A). A comparison of GSDensity-based gene set activity scores with pathway branch-level activity values from PSF showed similar spatial patterns of activation (Fig. 4C).

As with the clustering-based comparisons, we also observed a higher proportion of deregulated branches among the pathways shared with other methods (average ratio

of 0.394 significant sinks over total sinks in a pathway) compared to unique pathways identified only by PSF (average ratio of 0.268). Notably, PSF analysis identified nine additional spatially deregulated pathways not captured by GSDensity or spatialGE. These included the Hippo and Apoptosis signaling pathways, which showed strong spatial activation in the lower-right region of the tissue slice, a region corresponding to one of the most consistent clusters across all clustering methods based on both gene expression and pathway activity (Fig. S3). This cluster represents a highly proliferative melanoma region, which has also been previously described by *Schmidt et al. (2024)*. Another uniquely detected pathway was the ErbB signaling pathway, which showed increased activity in the upper-right corner of the tissue slice (Fig. S3). Notably, spatialGE identified nine pathways that were not detected with PSF nor with the GSDensity method.

In the mouse brain dataset, considerably less pathways were detected with spatialGE and GSdensity over spatial PSF (Fig. 4B). Two overlapping significant pathways were found between PSF and GSDensity method. Those two pathways showed similar activity patterns in mouse brain tissue between the methods (Fig. 4D).

## Pathway activity differences within clusters and interactions between adjacent cluster borders

An advantage of the PSF toolkit is its ability to calculate pathway activity at the branch level, which enables the exploration of potential communication between cells from different clusters. In particular, this allows us to investigate possible ligand–receptor interactions between deregulated pathways in adjacent tissue regions.

To demonstrate this capability, we examined how pathway activities varied within the same PSF-clusters based on their adjacency with other clusters in the melanoma dataset. We used the melanoma molecular subtype assignments (pigmentation-driven, proliferative, and immune-response subtypes) defined by *Schmidt et al. (2024)* to contextualize the differentially activated pathway branches observed in these border regions, linking spatial pathway changes to established tumor subtypes.

The analysis was performed only for Seurat-based PSF clusters with well-defined borders and sufficient spots in the cluster (PSF clusters 0, 1, and 3). According to the study by *Schmidt et al. (2024)* cluster 0 was characterized as melanoma type 1 with a pigmentation signature, cluster 1 as melanoma type 2 with an immune signature, and cluster 3 as proliferative melanoma.

For each of these clusters, we selected a group of spots from the core (surrounded by spots belonging to the same cluster) and one or more groups from the borders, adjacent to spots from other clusters (Fig. 5B).

The PSF-cluster 3, classified as proliferative melanoma, exhibited the fewest deregulated pathways between border and core spots. Notably, survival-associated pathways such as FoxO, Hedgehog, p53, ErbB, and cAMP signaling were significantly upregulated in the border spots adjacent to PSF-cluster 1, which may reflect interactions with immune cells that activate survival mechanisms in the proliferative melanoma cells (*Gajewski, Schreiber & Fu, 2013*).

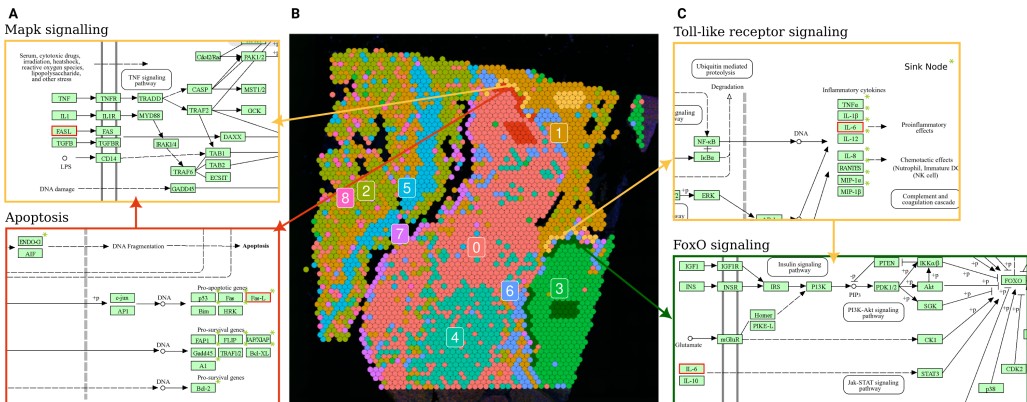

**Figure 5** **Terminal and input node connections between significantly deregulated pathways in border spot groups of adjacent clusters.** (A) A branch of the apoptosis pathway terminating in the FASL gene (outlined in red), upregulated in PSF cluster 0 (melanoma pigmentation), serves as an input node for the MAPK signaling pathway, which is upregulated in the adjacent PSF cluster 1 (melanoma immune response). (C) The Toll-like receptor pathway, activated in the border of PSF cluster 1, connects to FoxO signaling, activated in the border of PSF cluster 3, *via* the IL6 gene (outlined in red).

In PSF-cluster 1, several immune-related pathways were upregulated (Toll-like receptor signaling, chemokine signaling, Fc gamma R-mediated phagocytosis) in the border spots compared to the core of the same cluster. The border spots of PSF cluster 1 are adjacent to PSF-cluster 3, a highly proliferative cluster with activated Hippo signaling pathways. The proliferative nature of PSF cluster 3, driven by activated Hippo signaling, could contribute to the immune response in the border spots of PSF cluster 1, potentially through crosstalk between proliferative and immune signaling pathways in the microenvironment (*Yamauchi & Moroishi, 2019*).

Further exploration of terminal and input nodes of deregulated pathways in border spots of clusters 1 and 3 showed an interaction between the deregulated pathways of adjacent spot groups. In particular, one branch of the Toll-like receptor signaling pathway, which regulates the transcription of Interleukin 6 (IL-6), was significantly upregulated in the border spots of PSF-cluster 1 compared to its core. This cytokine activates the FOXO transcription factor through the Jak-Stat cascade in the FoxO signaling pathway, which is upregulated in the adjacent border spots of PSF-cluster 3 (Fig. 5C).

Comparison of border and core spot groups in PSF cluster 0 showed significant activation of the Apoptosis pathway. This may be linked to apoptosis-inducing signals originating from PSF cluster 1, which exhibited activation of immune response pathways in a group of spots adjacent to cluster 0 (Fig. 5B). The FASLG tumor necrosis factor, a terminal node in the apoptosis pathway, was activated in the border spots of the PSF-cluster 0 (melanocytes cluster). This factor also serves as a ligand in the MAPK signaling pathway, which was upregulated in the adjacent tissue from PSF-cluster 1, suggesting potential cross-talk between these pathways in adjacent tissue regions (Fig. 5A).

For the complete list of deregulated pathways in the border spot groups, see Dataset 3.

In summary, the analysis of differentially activated pathways within clusters revealed that border spots adjacent to other sections of cancer tissue may exhibit pathway activation or inhibition not observed in the cluster cores. Identifying these pathways and their potential connections between adjacent border groups can help uncover communication mechanisms between different tumor regions and further clarify how these interactions might contribute to treatment resistance, tumor progression, and metastasis.

### Spatial PSF browser

To facilitate exploration and analysis of pathway deregulation in spatial data, we developed the Spatial PSF Browser, an application designed to provide an interactive visualization of pathway activity profiles in spatial tissue slices in conjunction with pathway plots. Users can select a spot and a pathway, which is visualized by color-coded nodes (genes) based on gene expression or pathway activity values. One can also select individual pathway nodes to display their specific expression or PSF activity values in spots across the tissue slice. Additionally, a user can visualize the pathway activity based on the mean expression or PSF values of the selected cluster (Fig. S4). With this functionality, researchers can identify key genes or pathway branches that are activated in specific tissue sections or exhibit unique patterns of activation within the tumor microenvironment.

The spatial PSF browser was developed with the R Shiny framework, combining interactive spatial and pathway plots for dynamic exploration. It incorporates visualization Shiny modules from the PSF toolkit (*Hakobyan et al., 2023*) and the interactive spatial plot functionality. The PSF Spatial Browser with the integrated 10x Genomics melanoma spatial datasets is available https://apps.armlifebank.am/PSF_spatial_browser/. The spatial browser and pathway activity analysis functions for spatial datasets are integrated into the PSF toolkit R package and are accessible on GitHub (https://github.com/hakobyansiras/psf).

## DISCUSSION

In this study, we applied our topology-based pathway analysis tool, the PSF algorithm, to spatial transcriptomics datasets to investigate spatially localized signaling activity and functional heterogeneity within tissue sections. We used data generated with the 10x Genomics Visium platform, which captures transcriptomic profiles from spatially defined spots, each representing approximately ten cells. This resolution allowed the application of PSF without encountering data sparsity issues typically observed in single-cell transcriptomic datasets.

One of the main advantages of PSF over conventional gene set-based approaches (*e.g.*, ORA or GSEA) is its ability to generate continuous numeric activity values for each individual pathway branch. These values allow for unsupervised clustering and statistical testing at a branch level, offering a detailed and interpretable view of affected biological pathways.

In our previous study, we demonstrated that pathway activity analysis can be applied to single-cell transcriptomic data. The branch-level pathway activity can be used to cluster cells, aiding in the classification of cell types (*Hakobyan et al., 2023*). In this study, we applied both conventional clustering using the Louvain algorithm implemented in Seurat

and spatially aware clustering methods, including Vesalius and spatialGE on spatial transcriptomic datasets (*Stuart et al., 2019*; *Ospina et al., 2022*; *Martin et al., 2022*). We evaluated the clustering performance using both gene expression and pathway activity data from human melanoma and mouse brain spatial transcriptomics datasets. Both gene expression and pathway activity-based clustering showed low to moderate similarity across different clustering methods, with slightly higher values observed for gene expression-based clustering. This highlights the inherent difficulty of clustering spatial data and underscores the need for context-aware evaluation approaches. Pathway activity clustering provides a functional abstraction layer, reducing data dimensionality while highlighting biologically meaningful patterns. Second, because the PSF algorithm incorporates pathway structure, it can uncover functional groupings that are not apparent from expression profiles alone. This is particularly beneficial in complex tumor tissues, where the tumor microenvironment is composed of diverse, interacting cell types that are often difficult to distinguish using gene expression markers alone. In such cases, clustering based on shared functional states reflected in pathway activity profiles can more accurately capture biologically relevant groupings and reveal context-specific functional modules.

In addition to clustering, pathway activity values allow for more intuitive and biologically meaningful functional annotation. Cluster-specific features correspond to branches of signaling pathways that often represent key effector proteins involved in regulating specific cellular processes. The PSF toolkit also includes functionality such as partial influence analysis, which can be used to investigate the role of individual genes within pathway networks and to identify potential drivers of observed deregulation.

To evaluate the performance of our method in the context of spatial transcriptomics, we compared PSF-based analysis with established gene set-based methods. For clustering-dependent analysis, we used ORA on the marker genes of identified clusters. For clustering-independent spatial analysis, we applied the GSDensity and spatialGE methods which are the only avaialble methods performing spatially aware gene set analysis.

The spatialGE method uses simple statistics to identify spatially aggregated overexpression of genes for each gene set across the tissue map (*Ospina et al., 2022*). GSDensity is a more advanced method that uses multiple correspondence analysis to co-embed genes and cells into low dimensional space and then evaluate if the genes of the genesets have spatial coordination in that space. Then it calculates the relevance score of each gene-set for each spot/cell to use this data to identify spatially coordinated gene sets (*Liang et al., 2023*).

Because both PSF activity scores and GSDensity gene set relevance scores are continuous values with similar distributions, we applied the same kernel density estimation method from GSDensity to detect spatial specificity in our pathway activity data. Using this approach, we successfully identified nearly all of the spatially relevant pathways detected by GSDensity. Moreover, PSF analysis offered branch-level resolution, enabling the detection of spatially relevant pathway branches.

Another challenge in spatial transcriptomics analysis is the informative visualization of results concerning the tissue's spatial structure. The vast amount and diversity of results generated by spatial transcriptomics analysis necessitate user intervention and dynamic

exploration. Recently, *Schmidt et al. (2024)* developed a Spatial Transcriptomics Browser to aid interactive exploration of ST datasets. This platform provides a comprehensive environment for exploring ST datasets with Seurat as well as in-house (self-organizing maps; *Wirth et al., 2011*) clustering methods, and has an extensive collection of gene sets and visual tools for functional analysis and visualization. While this platform can be considered a significant improvement over existing ST visualization approaches, it lacks the instruments to visualize pathway deregulation, signal propagation, and altered individual pathway networks.

To facilitate further exploration of affected pathway branches and their potential impact, we have developed the PSF Spatial Browser. In addition to interactive visualization of expression and annotation information on the spatial tissue slice, the PSF Spatial Browser offers pathway activity and expression visualization directly on individual pathway networks, which provides a robust environment to explore and detect key genes and mechanisms associated with deregulation of those branches.

This study extends the applications of PSF analysis, demonstrating that topology-aware pathway analysis, in conjunction with the PSF toolkit's pathway curation and visualization tools, can be effectively applied to spatial transcriptomic datasets.

A notable limitation of PSF analysis on single-cell or high-resolution spatial datasets such as 10x Visium HD is data sparsity, particularly the prevalence of zero expression values can significantly reduce the effectiveness of PSF analysis, as many key genes within a pathway may be undetected, resulting in biased or incomplete activity estimates. In our previous work, we demonstrated that transforming sparse single-cell expression data into pseudo-bulk samples of 50 cells per group can improve the reliability of PSF-derived activity scores. Using the GTEx single-cell dataset, we further evaluated the relationship between the number of aggregated cells and the stability of PSF activity scores. We found that pathway activity estimates stabilize around the point where 10–12 cells are aggregated, which aligns well with the resolution provided by the $10\times$ Visium platform (Document S3).

Another limitation lies in the current availability of topology-rich pathways for analysis. For this study, we utilized gene expression fold change values of 2,097 genes from 40 pathways. While we could perform clustering with these 40 signaling pathways, many other pathways were not included in the analysis that could potentially be deregulated and play an important role in the analyzed datasets. The availability of high-quality pathways has always been a limiting factor for topology-based pathway analysis, as all the topology-rich pathways together cover only around 50% of human genes. Despite this, many consortia and research groups continue to study protein interactions and curate new pathways. With the help of pathway editing and parsing tools like the PSF toolkit, we can enlarge the collection of analyzed pathways and conduct more comprehensive analyses.

## CONCLUSIONS

We demonstrated the utility of topology-based gene expression analysis and visualization of spatial transcriptome data. This provides a new layer of information that complements and extends the traditional gene-centered analyses and can extend the studies of tissue

architecture and the cancer microenvironment at the pathway level. We also developed tools for visualizing and exploring pathway activity profiles in spatial data to facilitate systems-level analysis and interpretation.

## ACKNOWLEDGEMENTS

We would like to thank Ani Stepanyan for her invaluable contributions to the curation of KEGG signaling pathways, which greatly supported this study. We also extend our gratitude to Lilit Nersisyan for her insightful suggestions and engaging discussions. We also acknowledge the use of OpenAI's ChatGPT for assistance with language refinement and organization in the development of this manuscript.

### Funding

This work was funded by 21AG-1F021 and 22SC-BMBF-1C004 grants by the Committee of Higher Education and Sciences MESCS of Armenia. The paper is funded by the State Target Program of the Government of the Republic of Armenia under grant agreement No. 06-60-2025 project "Open Science National Cloud Infrastructure". The funders had no role in study design, data collection and analysis, decision to publish, or preparation of the manuscript.

### Grant Disclosures

The following grant information was disclosed by the authors:
The Committee of Higher Education and Sciences MESCS of Armenia: 21AG-1F021, 22SC-BMBF-1C004.
The State Target Program of the Government of the Republic of Armenia: 06-60-2025.

### Competing Interests

The authors declare there are no competing interests.

### Author Contributions

- Siras Hakobyan conceived and designed the experiments, performed the experiments, analyzed the data, prepared figures and/or tables, authored or reviewed drafts of the article, and approved the final draft.
- Maria Schmidt conceived and designed the experiments, authored or reviewed drafts of the article, and approved the final draft.
- Hans Binder conceived and designed the experiments, analyzed the data, authored or reviewed drafts of the article, and approved the final draft.
- Arsen Arakelyan conceived and designed the experiments, performed the experiments, analyzed the data, prepared figures and/or tables, authored or reviewed drafts of the article, and approved the final draft.
## Data Availability

The spatial transcriptomics datasets used in this study are publicly available on the 10x Genomics website; https://www.10xgenomics.com/datasets/human-melanoma-if-stained-ffpe-2-standard;

https://www.10xgenomics.com/datasets/mouse-brain-serial-section-1-sagittal-anterior-1-standard-1-1-0.

All code required to reproduce the results and figures are available at GitHub: https://github.com/hakobyansiras/psf/blob/main/inst/extdata/psf_spatial/psf_spatial_study_reproduction.R.

The PSF Spatial Browser with the integrated 10x Genomics melanoma spatial dataset can be accessed via https://apps.armlifebank.am/PSF_spatial_browser/.

The PSF Spatial Browser and spatial pathway analysis scripts are integrated into the PSF toolkit R package and are available at GitHub and Zenodo:

- https://github.com/hakobyansiras/psf.

- Hakobyan, S. (2025). PSF toolkit. In PeerJ (0.2). Zenodo. https://doi.org/10.5281/zenodo.15380893.

## Supplemental Information

Supplemental information for this article can be found online at http://dx.doi.org/10.7717/peerj.19729#supplemental-information.

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
