# Peer review of "Topology-aware pathway analysis of spatial transcriptomics"

_PeerJ, doi:10.7717/peerj.19729_

## Round 0.1 · original submission · Major Revisions

The reviewers' opinion is largely positive, but all see a room for significant improvement in the way the results are presented. The main problem seems to be that the authors have not decided whether they present a new tool with an example of analysis or a detailed case study supported by a new method, and this attempt to sit on two chairs simultaneously obscures the overall logic.

·

Basic reporting

Hakobyan et al. showcased the application of topology-aware pathway analysis to spatial transcriptomics (ST) data using the Pathway Signal Flow (PSF) toolkit. For their demonstration, they utilized melanoma ST data obtained from the 10x Genomics repository and developed an interactive web application for visualizing the results. While pathway analysis and visualization for ST data remain largely underexplored, these contributions offer valuable tools for the research community. However, the manuscript requires significant revisions to effectively highlight the authors’ contributions. In particular, the abstract and results sections are overly detailed with case study findings. The authors should succinctly emphasize the novelty of their contributions and streamline the presentation of results.

Additional comments:

1. [Margin line 19] “Understanding this spatial heterogeneity is essential for diseases such as cancer,...” " The use of “diseases” in the plural implies there should ideally be more than one disease listed. Consider specifying additional diseases or rephrasing to maintain consistency.

2. [Margin lines 29-46] If the authors aim to highlight the ‘topology-based pathway analysis approach’ as the main focus of the paper, detailing the findings from the case study in the abstract appears verbose and unnecessary. A concise summary of 2–3 sentences capturing the key findings would suffice.

3. [Margin line 58] “... particularly crucial in diseases such as cancer …”. Similar to line 19, “diseases” suggests more than one disease should be mentioned. Alternatively, rephrase for clarity.

4. [Margin line 63] “Although the resolution of ST is lower compared to single-cell transcriptomic approaches, this reduced resolution helps mitigate data sparsity issues that are commonly observed in single-cell transcriptomics, which can arise due to technical limitations of RNA extraction of small quantities.” The argument that spatial transcriptomics (ST) data mitigates data sparsity issues compared to single-cell transcriptomics lacks supportive evidence. Spatial transcriptomics datasets are generally more sparse. The authors should substantiate this claim by analyzing datasets from resources like the 10x Genomics website https://www.10xgenomics.com/datasets.

5. [Margin line 68] “One of the... analysis os the Seurat R package.” The statement regarding the limitations of Seurat’s analysis for ST data would be stronger if additional frameworks were discussed to illustrate the shortcomings of current tools.

6. [Margin line 70] “(Satija et al., 2015).” Since the paper focuses on ST data, the authors should cite the recent Seurat paper that introduced ST capabilities.

7. [Margin line 82] The claim “However, exploration of existing ST analysis…” is overly broad. It is not justified by examining only the Seurat framework. Consider tempering the statement or broadening the scope of analysis.

8. [Margin line 89]”... they are less sparse compared to single-cell datasets …” As previously mentioned, the claim that ST datasets are less sparse compared to single-cell datasets requires evidence.
[Margin line 92] “Recently, the spatial SOM browser …” The citation for the spatial SOM browser is missing.

9. [Margin line 113] “...SCT transformed…” On first use, the term “SCT” should be written in full to ensure clarity for readers unfamiliar with the abbreviation. Similarly, other terms are used in the manuscript.

10. [Margin line 186]”...spots of the image”. The phrase “spots of the image” should be revised to “spots of the tissue slide” or “spots of the sample” for better specificity.

11. [Figure 2] The figure caption should be revised for improved clarity. Labels like “Mel_type1” should be renamed to more descriptive terms, such as “Melanoma type 1”.

12. [Stratification of ST spots based on pathway activities] It’s not clear what the authors trying to highlight. The number of clusters detected by Seurat may change based on the hyperparameters of the Findclusters function. Authors must try clustering with varying hyperparameters to prove the analogy.

13. [Spatial PSF browser] The “Spatial PSF browser” is discussed in both the Materials and Methods and the Results sections with overlapping content. Consolidating the information into a single subsection under Materials and Methods would enhance clarity.

14. [Margin line 366] “... system-level analysis…” The term “system-level analysis” is ambiguous. Please clarify its meaning in this context.

15. [Margin line 369] “...(several hundreds of pathway… thousand genes)” The statement “(several hundreds of pathway… thousand genes)” is unclear. Rephrasing is recommended to improve readability and scientific rigor.

16. [Margin line 404] “...with multiple standard (Seurat)...” The term “standard” is vague, and multiple examples are expected, but only one is mentioned.

17. [Margin line 435] “...could perform sufficient clustering…” The term “sufficient clustering” lacks clarity.

Experimental design

no comment

Validity of the findings

To enhance reproducibility and transparency, the authors should deposit their code in an open data repository such as Zenodo or a similar platform.

·

Basic reporting

The authors present an analysis approach to study pathways in spatial transcriptomic data, more specifically using a method (PSF) conserving pathway topology. First, I would like to applaud the author for brining pathway analysis to the field of spatial transcriptomics. As they mentioned, there are few methods that provide a “pathway first” approach to spatial analysis. I believe that these types of tools are extremely valuable to the scientific community. In addition, the manuscript is clearly and concisely written.

While I acknowledge the importance of the work shown by the authors, I am not convinced that the paper in the current form does justice to their aim. To summarize, I am not sure if the paper is presenting a new method or a new analysis and in both cases I think there is room for improvement. I will consider this as method paper since this seems more likely. The work shown here does not provide a clear indication as to why the scientific community should consider their tool or their findings.

Experimental design

1. The authors mention the use of PSF toolkit that they had previously published. However, there is no mention of how the package was adapted to account for spatial context. The analysis pipeline shown in the paper follows a standard non-spatial analysis workflow using Seurat. If the goal is to demonstrate they have created a spatially aware approach to pathway analysis, I do not believe that this is what is shown in the manuscript. It is a pathway analysis tool which was applied to spatial data and contrasted to gene expression clusters provided by Seurat. If the goal was to demonstrate how pathways are spatial organized in melanoma samples, a single Visium 10X is not sufficient to make any biological conclusion.

For this point, the authors should clearly state which elements of their algorithm has been adapted to include a spatial component. This should be reflected in their analysis and in the manuscript. I suggest at the very least a demonstration of a non-spatial pathway method (even if it is their own PSF tool kit) in addition to downstream ORA or GSEA analysis. If it does not have a spatially ware component, then I am not sure that this counts a novel spatial method at all and is simply an analysis paper. In this case, as mentioned previously, a single Visium data set is not sufficient to provide trustworthy biological insights. The use of Seurat clusters in the pipeline should be replaced with spatially aware clustering method (SpatialPCA, BaseSpace, Vesalius, STARGATE, GraphST, etc)

It seems to me that the manuscript should encourage someone to use this tool as it will improve the insights they could gain from their data and how spatial transcriptomics needs adapted methods to get the most out of the data.


2. Building on the previous point, the GitHub shows that there are a couple of scripts that need to be “sourced” to load the required functions. Since the authors have already published PSF as a package, I would highly recommend adding these functions to that package. Or if that is not feasible, making this a separate and fully functional package (including manual pages, vignettes, internal data, etc). This package can use other tools to handle pre-processing as part of the pipeline. The current layout will not encourage people to use this package in their own pipeline. In addition, while Shiny apps are very useful for the broader biological community, unless the underlying tool has a solid backend and support, it becomes difficult to justify its use to the community.

3. There is a need to compare the method to other competing methods. The authors mention GSDensity and SpatialGE but do not show any comparison to these tools. I agree that that authors contrast their method with these tools by specifying that their method accounts for the topology of the pathway. It would be wise to demonstrate and contrast the performance of their tool to these other methods in the spatial context. Using pathway topology might improve biological insights in RNA-seq data but this might be trumped by methods which account for spatial context. I recommend looking as GSDensity, for instance, and using similar data sets (and additional tools) to quantitively benchmark the performance of the tool. The authors claim that topology aware adds another layer of information to traditional methods, but this is not reflected or clearly demonstrated in the in the manuscript.

To summarize the benchmarking aspect, I recommend that the authors quantitively and qualitatively demonstrate that their tools at the very least performs similarly to other spatial methods (GSDensity and SpatialGE) and outperforms non-spatial analysis methods (Seurat clusters, ORA, GSEA, and non-spatial PSF). For quantitative performance, the use of synthetic data is fine (the GSDensity paper provides real data examples as well). For qualitative performance, the authors should highlight what biological insights one could gain using their tool missed by other methods. While this might have been demonstrated in the authors previous publication, it has not been shown in spatial data hence the need for this paper.

4. I appreciate the depth in which the authors went for the biological interpretation of the Visium Melanoma sample. With that said, I fear that a single Visium data set might not best highlight the broad applicability of PSF. I would suggest adding at the least a couple more Visium data sets taken from different tissues and conditions. I think that multiple melanoma samples would also be acceptable since the authors might be able to highlight similarities or differences between patients. I would also add other spatial technologies such as MERFISH, CosMX, Xenium, Xenium 5k, Slide-seq, Stereo-seq or VisiumHD. I do not expect the authors to test all possible technologies, but I think that showing the use of the tool in other technologies (at least one image-based and one high resolution sequencing based) would strengthen the manuscript. If the tool is not able to handle these data sets, it should be clearly discussed.

Validity of the findings

No Comment

Additional comments

1. I would recommend avoiding using UMAP/tSNE plots if they are not required. In figure 1, for instance, they can either be removed or better yet replaced with heat maps.
2. Figure-1 and its caption is quite sparse in information and seems to show nearly the same information as in Figure 2. I recommend merging these figures together. Building on this point, both plots show Sankey plots from genes to pathways, but it would greatly aid if the pathways were more clearly annotated. What does P1 actually represent?
3. Overall, figure captions need to be expanded. A good rule of thumb is that a reader should be able to understand what the figure shows without referring to the main text. What is the key message that I should take away from this plot?
4. The supplementary figures show various cropped images from the online tool. It is good to show what it would look like, but I would recommend putting this into a single plot. For instance, a screen shot of the whole page, with boxes zooming into specific parts of the page explaining what it is. In a way, the authors did something similar in figure 5 where they show zoomed in boxes for different parts of the pathway. This would provide a more dense demonstration of their online tool.
5. The pathways are often complex to look at and when contrasting pathway activity between border and non-border, the authors mention connections between different parts of the pathway. While I recognize this might be a challenging task (feel free to ignore this comment), I would be nice to provide a way to automate the extraction of these links and connections instead of relying on what seems to be screenshots of the whole pathway.
6. Sometimes manual selection of points is sometimes unavoidable, in the case of spatial data, the selection of points on the border with other tissues is entirely possible and recommended for reproducibility. I would suggest adding a function to automatically create sub-clusters which represent border points. This can be achieve using KNN between 2 clusters (the RANN R package provides this with its nn2 function). Border points will be points which have a neighboring cluster in their nearest neighbors. Due to the hexagonal structure of Visium data, k=6 would yield neighboring spots only (expect spots on the edge of the tissue). It is also possible to achieve something similar using spatial proximity graphs.

Reviewer 3 ·

Basic reporting

The authors proposed the use of topology-aware PSF algorithm to assess and visualize pathway activity across tissue specimens. Using publicly available data of melanoma, authors attempt to compare and contrast the results between expression clusters (E-clusters) vs pathway activity analysis (P-clusters).

While English is clear and literature references were sufficiently cited, the current state of the manuscript can benefit from deeper analysis in supporting their claims.

The result paragraph from line 217 to 230 referes to results in Supplementary Fig.S4, a 5-by-8 bar charts, that are not well refered in the writing. Statements like "In contrast, the apoptosis pathway branches were mostly downregulated in this cluster suggesting that this part of the tissue slice is a highly aggressive and actively growing tumor (Supplementary Fig. S4)", were very hard to follow.

The authors also refered a number of times to Supplementary Data 1, which are very complex large 'raw data' although the authors make critical claims that PSF-cluster 1 showed 'immune-cold' phenotype. This is very hard to follow, and there appeared to be no quantitative comparisons, nor appropriate statistical comparison, among the different P-clusters.

The authors made similar claims throughout the result section: eg. line 251-263, claiming that cluster P2 showed upregulation of C-type lectin, requiring that the reader must compare 2300 lines of gene sets, without appropriate significance comparison among the different P-clusters.

Due to this equivocal claims, it is unfortunately impossible for readers to determine if their conclusion claim, "we observed substantial variability in the activity of key tumorigenesis-related pathways across different PSF clusters" was indeed accurate. To prove this point, proper statistic test, comparing gene set enrichment significance across all P-clusters are required, and the authors are authors suggested to take time in generating necessary figures (and subfigures) to walk the audience through these points carefully since it is one of the manuscript key claims.

The authors attempt to demonstrate the use of the Spatial PSF browser in Fig.5, comparing the borders vs core specimens. The authors are suggested to take time to defend the point, and walk the audience through this demo carefully. Supplementary Fig.S5,S6, S7 appeared to be quite crude in nature, without appropriate annotation.

Experimental design

-Lack appropriate statistical comparison when claiming that specific pathways are more dominant in specific P clusters.

Validity of the findings

-Most claims lack specific citations to evidence in figures.
-Supplementary data 1 must be reanalyzed and appropriately assessed to compare significance among the different P clusters, especially when defending for uniqueness of specific pathways.

---

## Round 0.2 · Minor Revisions

The reviewers have noted that the manuscript has been improved, but still have some comments, mainly editorial. The most serious problem, noted by both reviewers, is (non-)availability of the code.

·

Basic reporting

This version of the manuscript is greatly improved, and the authors also address many of my remarks. However, there remain some unproven claims and some places where the wording is unquantified. The author must address the following with regard to the current version:

1. This paper is mostly based on the data analysis and application of the Pathway Signal Flow (PSF) algorithm that the authors previously developed. The authors provide the GitHub repository; however, the repository does not contain a script to reproduce this analysis. The authors should include a reproducible script in the GitHub repository or place the script somewhere else that should be accessible to the readers.

2. Extension to the previous point. I could not run the examples provided in the GitHub readme.md successfully. It seems the example steps are incomplete.

3. [Line #261] “First, we compared clustering results obtained from gene expression and pathway activity data within each method for melanoma. The highest similarity between expression-based and pathway activity-based clustering was found for Vesalius (ARI of 57%), suggesting that this method better preserves cluster structure across data types compared to Seurat or spatialGE.” Comparison of Vesalius against Seurat and SpatialGE is not the primary purpose of this study. Moreover, stating that Vesalius preserves cluster structure across data types compared to Seurat or spatialGE, based on cluster comparison of gene expressions vs. PSF, may not be appropriate.

4. [Line #276] “The overall lower similarity in clustering results for the mouse brain dataset can likely be attributed to the lower median number of detected genes per spot in the mouse brain dataset compared to melanoma (6015 versus 7598 genes per spot, respectively). The lower gene detection rate in the mouse brain dataset may have negatively impacted the performance of pathway activity-based clustering, which relies on sufficient coverage of pathway-related genes.” No strong evidence is provided for this claim. To prove this argument, additional datasets should be analyzed, and trend lines can be plotted between the number of genes per spot and ARI.

5. [Lines #282–287] It is unclear whether the authors are referring to any figure in this paragraph. If not, an appropriate figure should be provided to support the claims made here.

6. [Line #289] “On the other hand, using PSF as input data produces clusters with consistency comparable to that of gene expression data.” No concrete value is provided as evidence.

7. [Line #303] “The results showed that PSF analysis identified a substantially higher number of significant pathways in all clusters compared to ORA and GSDensity.” The authors should refrain from using terms such as “significant” without statistical testing to support the claim.

8. [Line #509] “All three clustering approaches yielded broadly consistent results across both expression and pathway activity data types, supporting the use of PSF activities as a suitable input for single-cell and spatial data analysis.” This contradicts what was previously stated in Lines #261–281.

Experimental design

NA

Validity of the findings

NA

Additional comments

NA

·

Basic reporting

First, I would like to thank the authors for their work and the extensive changes they have made to the manuscript. I better understand how the method works with respect to the spatial component. It produces value that can be used for clustering (which can be spatially aware clustering). I hope that this version of the manuscript will have the impact the work deserves.

There are few minor aspects that I would like to point out for the sake of clarity.

1. I thank the authors on expanding the figure captions. I would only ask that the authors ensure that their labelling is consistent. For instance, instead of using “Panel D”, using the same nomenclature as for the other panels. (D). Along these lines, please make sure that all figures are referenced within text and in a logical order (A B C D …). While this is minor it does help the reader to follow the flow of the story with more ease.
2. Overall, the manuscript is clearly written, and I appreciate the authors openness about the use of LLMs. There are a few sentences that to me are not clear as to what the authors are trying to say. A slight reformulation could make the message a bit clearer. They are listed below:
34-36 “PSF-based clusters showed good correspondence with expression clusters with spatially aware
clustering methods.”
43-45 “We further demonstrated that PSF-derived activity values can be used to detect spatially deregulated pathways similarly to spatially-aware gene-set enrichment
estimating approaches GSDensity and spatialGE.”

Experimental design

I would suggest adding the link the analysis to the paper as well. It was present in the rebuttal but not in the paper itself. The code seems like it could be run as is but it is not easy to find.

Validity of the findings

No Comment

---

## Round 0.3 · accepted · Accept

The reviewers and the editor are satisfied with the final editorial changes.

·

Basic reporting

I would like to commend the authors for their diligent effort in addressing the suggestions provided during the review process. The revisions have significantly enhanced the clarity and quality of the manuscript.

At this stage, I do not have any additional comments or suggestions on the manuscript. I believe the manuscript is well-prepared and ready for consideration in its current form.

Experimental design

None

Validity of the findings

None